# A Bayesian hierarchical model of trial-to-trial fluctuations in decision criterion

**Robin Vloeberghs**[1]*, **Anne E. Urai**[2], **Kobe Desender**[1‡], **Scott W. Linderman**[3*‡]

**1** Brain and Cognition, KU Leuven, Leuven, Belgium, **2** Cognitive Psychology, Leiden University, Leiden, The Netherlands, **3** Department of Statistics and Wu Tsai Neurosciences Institute, Stanford University, Stanford, California, United States of America

‡ Shared last authorship.
* robin.vloeberghs@kuleuven.be (RV); scott.linderman@stanford.edu (SWL)

**Data availability statement:** All relevant data are within the manuscript and its Supporting information files. A comprehensive demo and

## Abstract

Classical decision models assume that the parameters giving rise to choice behavior are stable, yet emerging research suggests these parameters may fluctuate over time. Such fluctuations, observed in neural activity and behavioral strategies, have significant implications for understanding decision-making processes. However, empirical studies on fluctuating human decision-making strategies have been limited due to the extensive data requirements for estimating these fluctuations. Here, we introduce hMFC (Hierarchical Model for Fluctuations in Criterion), a Bayesian framework designed to estimate slow fluctuations in the decision criterion from limited data. We first showcase the importance of considering fluctuations in decision criterion: incorrectly assuming a stable criterion gives rise to apparent history effects and underestimates perceptual sensitivity. We then present a hierarchical estimation procedure capable of reliably recovering the underlying state of the fluctuating decision criterion with as few as 500 trials per participant, offering a robust tool for researchers with typical human datasets. Critically, hMFC does not only accurately recover the state of the underlying decision criterion, it also effectively deals with the confounds caused by criterion fluctuations. Lastly, we provide code and a comprehensive demo to enable widespread application of hMFC in decision-making research.

## Author summary

Every day, we make numerous decisions, from choosing what to eat to how we interpret the world around us. Traditionally, researchers have assumed that a key part of our decision-making process, namely the decision criterion, stays stable over time. However, increasing evidence suggests that instead of being stable, the decision criterion can fluctuate from moment to moment. In this work, we introduce the Hierarchical Model

code can be found at
www.github.com/robinvloeberghs/hMFC.

**Funding:** AEU is supported by a Veni fellowship
(VI.Veni.212.184) from the Netherlands
Organisation for Scientific Research. SWL is
supported by grants from the NIH BRAIN
Initiative (U19NS113201, R01NS131987,
R01NS113119, & RF1MH133778), the NSF/NIH
CRCNS Program (R01NS130789), and the
Sloan, Simons, and McKnight Foundations. The
funders had no role in study design, data
collection and analysis, decision to publish, or
preparation of the manuscript.

**Competing interests:** The authors have
declared that no competing interests exist.

for Fluctuations in Criterion (hMFC), a new computational model that can accurately estimate these fluctuations, even with limited data. Capturing these moment-to-moment changes is critical: ignoring them can bias estimates of perceptual sensitivity and the influence of past choices on current decisions. By estimating criterion fluctuations, hMFC can correct these biases. With hMFC, we offer researchers a powerful tool for uncovering the dynamic nature of human decisions. To support wide adoption, we also provide open source software together with a demo, making this approach accessible to the broader scientific community.

## Introduction

Every day we make a multitude of decisions: deciding when to cross the street, whether to add a newly discovered song to your playlist, or choosing between coffee or tea. Much effort has gone toward understanding the computational mechanisms underlying perceptual [1], value-based [2], risky [3], and social decisions [4]. A common theme in these frameworks is that decisions are formed by comparing evidence to a decision criterion. Classical models of decision-making assume that observers generate an internal representation of the relevant information, typically referred to as "evidence" or the "decision variable" [5]. To generate a binary choice, this decision variable is compared to an internal decision criterion. Due to the inherent noisiness of the brain or changes in internal states (e.g., changes in attention), repeated presentation of the same stimulus will not always lead to the same internal representation of that stimulus [6,7]. Instead, multiple trials of the same stimulus produce a distribution of internal decision variables (Fig 1A). This inherent variability explains why participants make variable responses when repeatedly presented with the same stimulus.

A key component in these models is the decision criterion, which allows to explain biases in decision-making. When observers choose one answer more often than another, their behavior can be modelled as the result of a shift in decision criterion, independent of their sensitivity to decision-relevant information [5]. Empirically, the decision criterion depends on internal and external factors: for instance, it shifts in response to external feedback [8], expectancy [9], task instructions [10], monetary rewards [11], and the base rate of stimuli [12].

Although it is widely recognized that the decision criterion can be flexibly shifted depending on the environment, a common implicit assumption is that in the absence of experimental manipulations, this (potentially biased) criterion remains constant across a series of experimental trials. Although this assumption allows computational models to be simple and tractable, it may be overly simplistic. Indeed, the idea of trial-by-trial variability in the criterion can be traced back to early work [13–15], and has been noted often in the literature [16,17]. However, this line of work focuses mostly on the criterion distribution (mean and variance), rather than attempting to estimate the trial-to-trial trajectory of the fluctuating decision criterion.

More recently, the dynamical nature of cognitive processes has received renewed interest in cognitive and computational neuroscience [18]. One reason for this trend is the availability of new computational tools to quantify trial-by-trial fluctuations from behavioral and/or neural data. Indeed, increasing evidence suggests that several computational parameters that are thought to underlie decision-making are not stationary, but rather fluctuate over trials [19–23]. For example, Ashwood and colleagues [19] demonstrated that decision behavior in mice can be described by three discrete states persisting for 10 to 100 trials, each with different psychometric parameters. Other work has focused on modeling continuous trial-by-trial

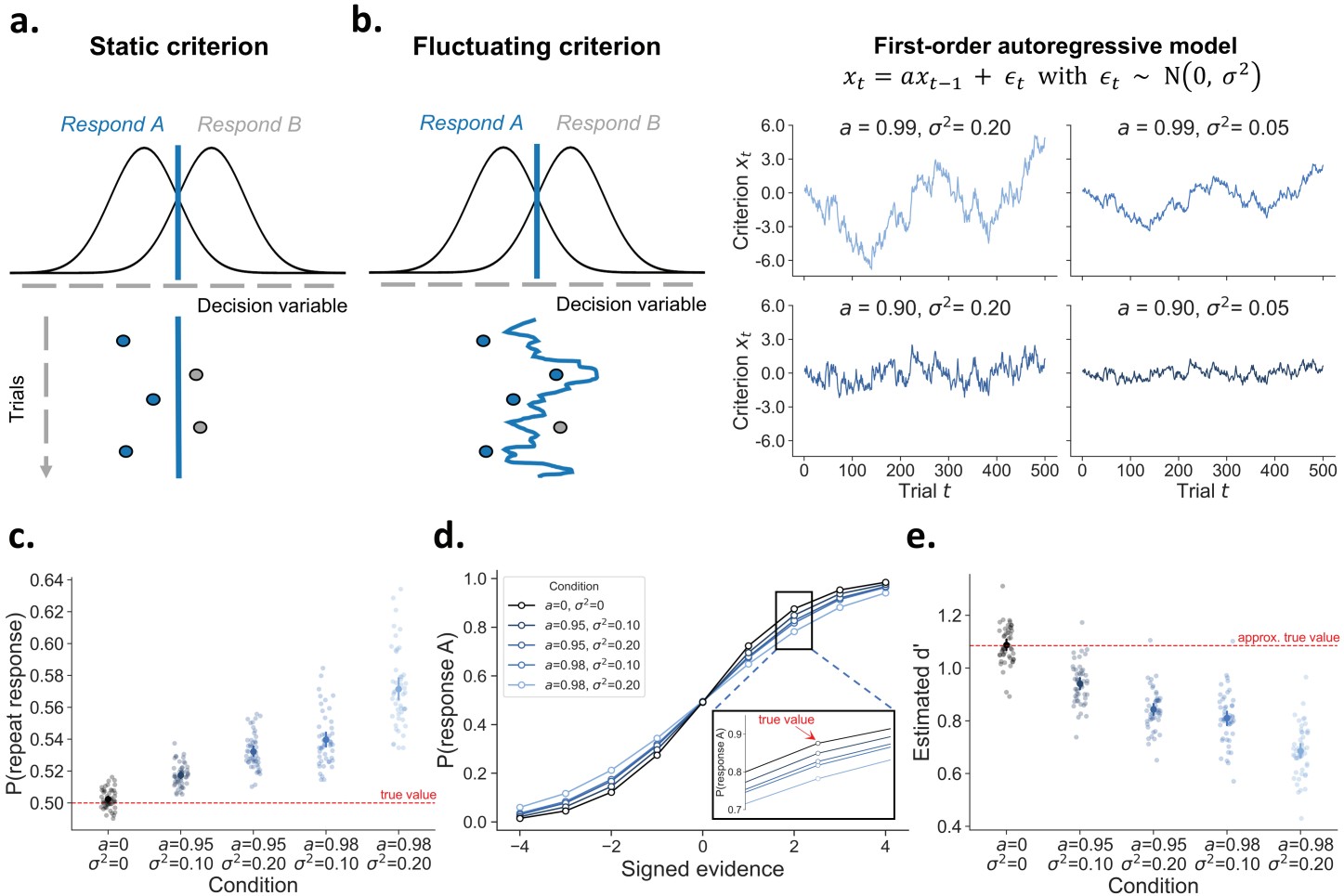

**Fig 1. The consequences of fluctuations in decision criterion. A)** Many theories of decision-making assume that choices are formed by comparing a decision variable to a criterion. According to signal detection theory, an observer will respond A or B, depending on whether the decision variable falls left or right to this criterion, respectively. Whereas this criterion is typically assumed to be fixed across a series of trials (static criterion), here we investigate the consequences of this criterion slowly fluctuating across time (fluctuating criterion). **B)** Fluctuations in criterion can be modelled as a time series using a first-order autoregressive model AR(1). An AR(1) model has two free parameters that control the temporal dependency ($a$) and the scale of the fluctuations ($\sigma^2$). The four panels show how the criterion changes over time under different values of these two parameters. **C)** Simulations show that it is of critical importance to consider fluctuations in decision criterion. Despite the absence of an effect of previous response (true $w_{\text{previous response}} = 0$), they give rise to artificial choice history effects. **D)** Likewise, while the generative psychometric slope (true $w_{\text{stimulus}} = 1.25$) is constant for all conditions, fluctuations in decision criterion lead to an underestimation of the slope of the psychometric function. **E)** Fluctuations in criterion lead to an underestimation of $d'$, a popular signal-detection measure of sensitivity.

changes in the overall bias (i.e., intercept; [21]) and other parameters of a logistic regression model [23]. Combining behavior with neural recordings, Cowley et al. [20] found that during a multi-hour session, monkeys' hit and false alarm rate slowly drifted (i.e. suggesting a slow drift in the decision criterion) and co-varied with neural activity in area V4 and prefrontal cortex. In other work, Mochol and colleagues [22] showed that decision behavior in monkeys is influenced by fast (previous trial's response and reward) and slow (moving window of 130 trials) biases, both encoded in the PFC, that are integrated into the current decision.

Despite much progress in this line of research, an important shortcoming in the current literature is that the few methods that are available to estimate the decision criterion at the *single-trial* level require too many trials and fix crucial parameters instead of estimating them.

In this work, we propose a new model that solves these issues. First, we start with investigating the consequences of fluctuations in decision criterion. We do so by simulating the effects of fluctuations in criterion on common measures of decision-making behavior and highlight the problems that these fluctuations may cause in interpreting experimental data. We then present a novel hierarchical framework that allows trial-to-trial estimation of fluctuations in the decision-criterion and show that its parameters can be recovered even with a small number of trials (i.e., 500). Lastly, we show that this model solves the confounds caused by unaccounted fluctuations in decision criterion.

## Why should we care about fluctuations in decision criterion?

In the next section we simulate the empirical consequences of criterion fluctuations, and especially highlight how these could impact different experimental conclusions. Rather than assuming a fixed decision criterion across time, we model the criterion fluctuations as a first-order autoregressive process (AR(1); see [21] and [23] for similar approaches). An AR(1) model describes the temporal dynamics of a time series, in this case the decision criterion, via two parameters. First, an autoregressive coefficient $a$ captures the temporal dependency between two successive time points ($x_t$ and $x_{t-1}$). In general, for an AR(1) model to have stable behavior, $a$ must take values between -1 and +1. When $a = 0$, the resulting criterion is sampled from a normal distribution without across-trial autocorrelation. Second, a zero-mean error term with variance $\sigma^2$ causes the random fluctuations. Fig 1B illustrates how variations in both these parameters influence a simulated time series of the decision criterion. Smaller magnitudes of $a$ (bottom left panel) make a time series more mean-reverting, with shorter fluctuations around its mean. In contrast, when a time series is less mean-reverting due to larger values for $a$ (top left panel), it will deviate more strongly from its mean for longer periods. The variance of the error term $\sigma^2$ governs the scale of the random fluctuations and thus acts as a scaling parameter, as can be seen by comparing the left and right panels. The left and right panel time series show the exact same dynamics but merely differ in magnitude on the $y$-axis.

Having established a model to generate fluctuations in decision criterion, we next use simulations to show that these fluctuations can lead to biased and erroneous inferences in various commonly used metrics that (implicitly) assume stable criteria. Below, we discuss three examples highlighting the importance of accounting for a time-varying criterion. First, the presence of criterion fluctuations can mimic a causal influence of previous responses, causing apparent sequential effects in decision-making. Second, criterion fluctuations lead to biased estimates of popular measures of sensitivity. Specifically, both the slope of the psychometric function and $d'$ from SDT are underestimated in the presence of criterion fluctuations. Third, the ability to quantify criterion fluctuations from trial to trial will contribute to a better understanding of decision-making and its neural basis.

**Criterion fluctuations can create apparent sequential effects.**  A wealth of studies have suggested that decisions are not isolated events, but instead show sequential effects: past trials bias the current decision. Such sequential effects have been observed in humans [24–32], primates [22,33], and rodents [21,34–37] for a wide range of tasks. In 2AFC tasks, one example of a sequential effect is the tendency to repeat previous responses, also known as choice history bias, repetition bias, serial dependencies, perseveration, or hysteresis [25,30,31,38–44]. These effects are often taken as evidence that observers maintain active representations of their environment or decision history: an influence of previous responses is typically interpreted and modeled as causal effect, where observers update their criterion based on some combination of past stimuli, choices and rewards [24,30,34,43]. For example, Treisman and

Williams [15] suggested that sequential effects can be explained by adjusting the criterion on the current trial as a function of the observer's previous response.

However, sequential effects can also arise in absence of any "systematic" adjustments, purely as the consequence of random fluctuations in the decision criterion [see also, 21, 34,45]. To illustrate this point, we simulate five different datasets consisting of 50 agents with 5000 trials each. Critically, we simulate behavior in which the decision criterion is not stable but instead fluctuates over time according to a first-order autoregressive process. To alter the dynamics of the criterion fluctuations, each dataset is generated with different values for $a$ and $\sigma^2$. On each trial, the response is drawn from a Bernoulli distribution with the probability governed by stimulus evidence (true $w_{stimulus} = 1.25$), previous response (true $w_{previous\ response} = 0$), and a fluctuating value over trials representing the fluctuations in criterion. Note that adding this fluctuating value to the Bernoulli probability is mathematically identical to comparing a decision variable to a varying criterion (i.e., fluctuating criterion; Fig 1A). When this value does not fluctuate ($a = 0$ and $\sigma^2 = 0$) this corresponds to a fixed and unbiased criterion (i.e., static criterion; Fig 1A). In Fig 1C, we confirm that simulated observers using a fixed criterion do not display a tendency to repeat the previous response (i.e., no choice history bias). However, a fluctuating decision criterion causes the emergence of *apparent* choice history bias, despite the absence of any "systematic" criterion shifting in function of previous response (true $w_{previous\ response} = 0$) (see [34, Fig 2], and [21] for similar simulations). As expected, this apparent choice history bias increases when the random fluctuations in decision criterion become stronger (i.e., increased $a$ and $\sigma^2$). In sum, researchers interested in choice history effects cannot ignore the possibility of fluctuations in the decision criterion.

**Criterion fluctuations lead to an underestimation of perceptual sensitivity metrics.** Many researchers are interested in perceptual sensitivity, which is often measured using one of two popular methods: the psychometric slope and $d'$ in signal detection theory (SDT). Both assess how well individuals can discriminate stimuli, and each has its own method of quantifying sensitivity. The psychometric slope is obtained by presenting stimuli at multiple levels of intensity and fitting a psychometric function to the data. The slope from this function is often directly used as a measure of sensitivity, or researchers may infer a threshold of interest by evaluating the slope at a particular stimulus intensity strength [46]. However, in the presence of criterion fluctuations, the slope of the psychometric function is severely biased (Fig 1D). Specifically, the estimated slope of the psychometric function decreases with increasing strength of criterion fluctuations, despite having the same generative psychometric slope (true $w_{stimulus} = 1.25$). This underestimation occurs because fluctuations in the criterion lead to more variable responding to identical stimuli, which flattens the psychometric function [16,27,47]. Such distortions may impact scientific conclusions, for example when determining the threshold for conscious access [48], or a specific level of performance [49]. In sum, researchers applying psychometric functions in their work should be aware of criterion fluctuations and its consequences.

Similarly, inspired by signal detection theory researchers often use $d'$, the difference between z-scored hits and z-scored false alarms, as a measure of sensitivity [47]. For a given level of true sensitivity, $d'$ remains unchanged in the presence of biased decision criteria (cf. iso-sensitivity curves; [5]). Although this property holds in the presence of a fixed response bias (i.e., static criterion; Fig 1A), the property does not hold when the criterion fluctuates [13]. Specifically, criterion fluctuations lead to underestimation of the true sensitivity, biasing estimates of $d'$. To demonstrate this point, we again simulate behavior for different observers that all have the same true sensitivity (i.e., same $w_{stimulus}$), but differ in criterion

fluctuation dynamics. The simulation follows the same procedure as before (Fig 1D) but now only includes one level of (signed) evidence. If we compute $d'$ based on the data without criterion fluctuations, we get an estimated $d'$ of 1.09 (Fig 1E). However, while the average criterion is accurately recovered (S1 Fig), the presence of criterion fluctuations biases $d'$ estimates downwards. In sum, researchers interested in comparing sensitivity, as measured by $d'$, across participants or conditions should account for the possibility of criterion fluctuations.

**Criterion fluctuations as a topic of study in human neuroscience.** The above examples illustrate how criterion fluctuations can result in various biases. Note that the impact of fluctuations is not limited to only these examples: the concern that non-stationarities might underlie post-error and post-confidence slowing—i.e., slow fluctuation in the decision boundary of the drift diffusion model—has been raised in other domains as well [50,51]. While these examples stress the importance of controlling for these fluctuations, it is crucial to highlight that the criterion fluctuations are themselves an intriguing phenomenon worthy of exploration. A growing body of research illustrates that decision-making is characterized by non-stationarities [19–23,50–53]. Whereas these non-stationarities seem to be a fundamental feature of decision-making, they are still often ignored or considered a nuisance. Recently, a general trend in cognitive neuroscience is emerging towards more dynamical approaches, with more effort spent to capture time-varying internal states [7,18,54]. The ability to measure trial-to-trial criterion fluctuations opens up new avenues for research, such as uncovering the neural mechanisms underpinning them, identifying more phenomena in which they play a role, and leveraging this understanding to improve computational models. For example, quantifying criterion fluctuations with an AR(1) model and estimating its parameters allows the comparison of these parameters over subjects and experimental manipulations. In sum, studying criterion fluctuations will advance our understanding of decision-making and its underlying mechanisms.

## Previous attempts to measure criterion fluctuations

In the previous section, we showcase different scenarios illustrating the importance of considering fluctuations in the decision criterion. In the context of choice history bias (i.e., repetition effects in function of previous response), earlier work attempted to control for criterion fluctuations through a model-free approach [34,45], similar to one developed in the post-error slowing literature [50]. Since the criterion is assumed to be autocorrelated, it should have a similar influence on temporally adjacent trials. In theory, it should thus be possible to dissociate: (i) a causal influence of the previous response on the current trial, from (ii) an acausal influence of the next response on the current trial, which reflects the value of the criterion that varies only slowly across several trials. Lak et al. [34] proposed to obtain a drift-free measure of choice history bias by subtracting the acausal from the causal influence. Although this model-free approach sounds promising, Gupta and Brody [21] demonstrated that this is only effective in the context of a specific subset of updating strategies (i.e., no systematic updating and symmetric win-stay lose-switch). In the presence of other updating strategies, such as win-stay lose-stay or win-stay lose-nothing, their model-free correction resulted in inaccurate and biased estimates. Therefore, this proposed model-free approach is not an appropriate tool to correct for criterion fluctuations. Moreover, it does not allow extracting a trial-by-trial measure of the latent criterion to study its neural and physiological basis.

Instead of the model-free approach, Gupta and Brody [21] have proposed a model-based approach which explicitly models criterion fluctuations as an AR(1) process in a Bernoulli

linear dynamical system. With this approach they were able to correctly recover a wide range of updating strategies. However, there are some key limitations that prevent a widespread usage of their approach, especially in human cognitive neuroscience. First, the authors only tested their model using very large datasets: simulated datasets with 40.000 trials, or rat data with 50.000 trials, obtained over many sessions. Second, the authors only focused on the recovery of history biases but did not quantitatively investigate the recovery of the criterion trajectory and its underlying computational parameters. Instead, they assumed a fixed autoregressive coefficient a of .9995. Similarly, Roy et al. [23] developed a method to estimate trial-by-trial fluctuations in the weights of a logistic regression model, which captures criterion fluctuations through the intercept. However, the weights were assumed to evolve over time according to an AR(1) process with *a* fixed to 1 (i.e., a random walk). Given that the parameter *a* plays a crucial role in capturing the temporal dynamics of a time series (i.e., as shown in Fig 1B), fixing this parameter to a specific value is therefore an overly strong assumption. It is likely that the generative autoregressive coefficient of criterion fluctuations in experimental data deviates from this value, which may bias estimates of trial-to-trial criterion trajectories.

## Hierarchical Model for Fluctuations in Criterion (hMFC)

In the current work, we develop the Hierarchical Model for Fluctuations in Criterion (hMFC), which aims to solve these limitations by extending earlier approaches in various respects. First, to solve the issue that the model requires a high number of trials to be fitted, we implement a Bayesian hierarchical estimation approach. This allows to combine data across multiple participants and concurrently models both group level estimates and individual estimates. Therefore, fewer trials are required to obtain reliable parameter estimates. Second, instead of fixing *a* in the AR(1) to .9995 [21] or 1 [23], we implement this in our model as a free parameter. This is of crucial importance because when the true value for *a* deviates from the assumed value, it causes biases in the parameters estimates and a poorer recovery of the criterion fluctuations, as will be illustrated later using empirical and simulating data. Having *a* as a free parameter does require a hierarchical estimation approach for good recovery at lower trial numbers (S2 Fig). This free parameter allows to capture a wide range of criterion fluctuation dynamics and ultimately enables researchers not just to control for criterion fluctuations, but also to study these fluctuations and its dynamics as the main topic of investigation (e.g., comparing a over different experimental manipulations or investigating individual differences). Finally, we provide easy to use code and software demos to enable the application of our model by the cognitive neuroscience community.

In sum, in order to model fluctuations in the decision criterion, we implement a hierarchical framework within a Bernoulli linear dynamical system, called the Hierarchical Model for Fluctuations in Criterion (hMFC). Next, we elaborate on the technical details of hMFC.

## Methods

### The generative model

First, consider a single session consisting of a sequence of $T$ trials. On each trial $t = 1, \ldots, T$, the agent makes a binary response $y_t$. Our generative model assumes the response is drawn from a Bernoulli distribution:

$$y_t \sim \mathrm{Bern}(f(\boldsymbol{w}^\top \boldsymbol{u}_t + x_t)) \tag{1}$$

where the probability of a response is governed by the covariates $\boldsymbol{u}_t \in \mathbb{R}^p$, the latent fluctuating criterion $x_t \in \mathbb{R}$, and the weights $\boldsymbol{w} \in \mathbb{R}^p$. The sigmoid transformation, $f(z) = 1/(1 + e^{-z})$,

constrains the conditional probability to be between 0 and 1. The vector $\boldsymbol{u}_t$ contains the observed input variables or covariates on trial $t$, such as the stimulus evidence or previous response. The influence of these covariates on the Bernoulli probability is captured by the weights $\boldsymbol{w} \in \mathbb{R}^p$. The fluctuating criterion $x_t \in \mathbb{R}$ on trial $t$, which is not directly observable, is assumed to follow a first-order autoregressive process:

$$x_t = b + a x_{t-1} + \epsilon_t, \qquad \epsilon_t \sim \mathrm{N}(0, \sigma^2) \tag{2}$$

where $a \in [0, 1]$ is the autoregressive coefficient which captures the temporal dynamics of the criterion fluctuations (Fig 1B), and $\sigma^2$ controls the variance of the fluctuations (i.e., $\sigma^2$ is a scaling parameter). The intercept $b$ enables to estimate the mean of the criterion trajectory, which in turn allows to capture overall biases. Note that adding a drifting value $x_t$ to the log-odds of the Bernoulli distribution is mathematically equivalent to comparing a decision variable to a fluctuating criterion (Fig 1A). In sum, the model predicts responses based on a combination of the weighted variables $u_t$ and the fluctuating criterion $x_t$, as illustrated in Fig 2A. Ultimately, this model allows us to disentangle two sources that influence the response probability: (i) the influence of $\boldsymbol{u}_t$, and (ii) the influence of the fluctuating criterion $x_t$.

## The hierarchical prior: putting the 'h' in hMFC

Having explained the generative model and the behavior it generates (Fig 2A), we next focus on the inverse step, namely estimating the underlying latent parameters given observed behavior. In many settings, we need to obtain accurate parameter estimates from only tens to hundreds of trials per subject. Small values of $T$ pose statistical challenges for parameter estimation. However, we can pool information across individuals to aid in inference. To this end, we developed a hierarchical Bayesian model that allows for per-subject parameter estimates, while sharing information across subjects via a global prior distribution (Fig 2B). We infer the per-subject parameters, the latent criterion trajectory for each subject, as well as the strength of the hierarchical prior using a fully Bayesian inference algorithm.

In the hierarchical model, each subject $i = 1, \dots, N$ has its own responses $y_{i,t}$, covariates $u_{i,t}$, and fluctuating criterion, $x_{i,t}$, for $t = 1, \dots, T_i$. Each subject also has its own parameters $\theta_i = (\boldsymbol{w}_i, a_i, \sigma_i^2, \mu_{x,i})$, where $\boldsymbol{w}_i \in \mathbb{R}^p$ denote the covariate weights, $a_i \in [0, 1]$ is the autoregressive parameter, $\sigma_i^2$ is the variance of the latent criterion fluctuations, and $\mu_{x,i}$ is the mean of the criterion trajectory. Using the estimates for $a_i$ and $\mu_{x,i}$ the AR(1) intercept $b_i$ (Fig 2B) is derived as $b_i = \mu_{x,i}(1 - a_i)$. We learn a hierarchical prior that captures how the subject-level parameters $\theta_i$ vary across the population. The hierarchical prior allows us to share statistical power from other subjects when estimating $\theta_i$. This results in better parameter recovery, even when a relatively small number trials per subject is available [55].

We specified hierarchical priors for the four types of per-subject parameters. First, we assume a Gaussian prior on the weights,

$$\boldsymbol{w}_i \sim \mathrm{N}(\mu_{\boldsymbol{w}}, \mathrm{diag}(\sigma_{\boldsymbol{w}}^2)), \tag{3}$$

where $\mu_{\boldsymbol{w}} \in \mathbb{R}^p$ and $\sigma_{\boldsymbol{w}}^2 \in \mathbb{R}_+^p$ denote the global mean and variance of the per-subject weights. Second, we assume a truncated normal prior on the per-subject autoregressive coefficient,

$$a_i \sim \mathrm{TruncNorm}(\mu_a, \sigma_a^2, [0, 1]) \tag{4}$$

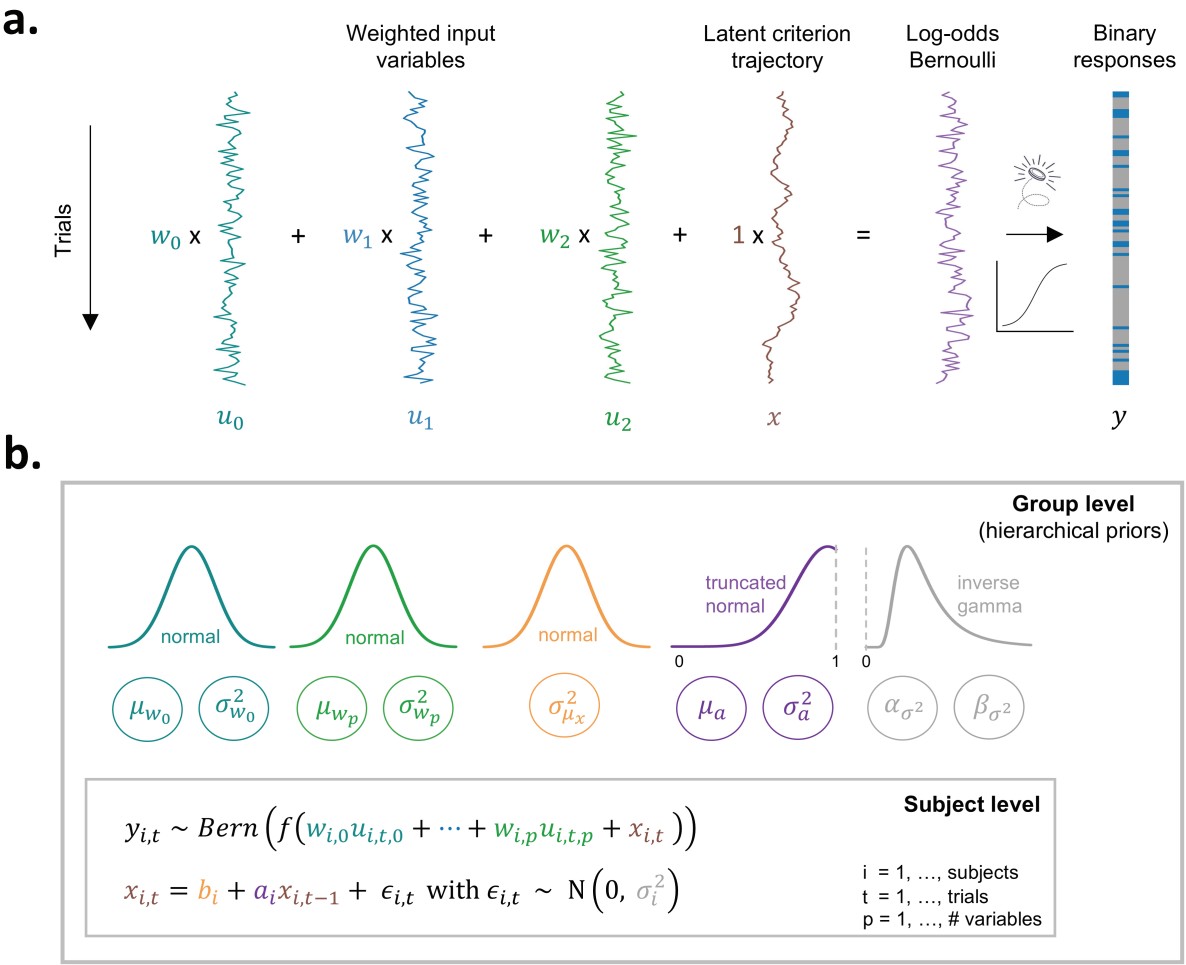

**Fig 2. An overview of the generative model and its hierarchical structure. A)** The figure displays, using 100 simulated trials for one agent, responses sampled from a Bernoulli distribution (i.e., a series of weighted coin flips). The trial-by-trial Bernoulli probabilities are a function of a weighted combination of the observed covariates $\boldsymbol{u}_t$ and the latent criterion fluctuations $x_t$. The first three columns show how the covariates, drawn from a standard normal distribution $N(0, 1)$ evolve randomly over time. Summing the weighted covariates with the criterion trajectory results in the log-odds of the Bernoulli distribution. After applying a sigmoid transformation to transform log-odds into probabilities, we draw from the Bernoulli distribution (i.e., weighted coin flip) to produce binary responses. **B)** For each subject $i$ we infer weights $(w_{i,0}, w_{i,1}, \ldots, w_{i,p})$ and model the latent criterion fluctuations $x_{i,t}$ as an AR(1) process with per-subject intercept $b_i$, autoregressive coefficient $a_i$, and variance $\sigma_i^2$. Through the specification of hierarchical priors we allow the sharing of statistical strength across subjects when estimating these individual parameters. Like the individual parameters, the parameters of the hierarchical distributions are iteratively updated during the estimation procedure. Note that $b_i$ is not estimated directly and therefore does not have a hierarchical prior. Instead, we estimate the mean of the criterion trajectory $\mu_{x,i}$, for which a normal hierarchical prior with zero mean and $\sigma_{\mu_x}^2$ as variance is assumed. Following the formula for the mean of an AR(1) process we can derive $b_i = \mu_{x,i}(1 - a_i)$.

where $\mu_a$ and $\sigma_a^2$ are the global mean and variance, respectively, and the domain is constrained to [0,1] because negative values of $a_i$ would lead to large jumps in the criterion, and values greater than 1 would lead to unstable dynamics. Third, we specify an inverse gamma (IGa) prior for the variance of the criterion fluctuations,

$$\sigma_i^2 \sim \text{IGa}(\alpha_{\sigma^2}, \beta_{\sigma^2}), \tag{5}$$

with shape parameter $\alpha_{\sigma^2} \in \mathbb{R}_+$ and a scale parameter $\beta_{\sigma^2} \in \mathbb{R}_+$. Note that $\alpha_{\sigma^2}$ is not estimated directly. Instead, we estimate $\mu_{\sigma^2}$ and derive $\alpha_{\sigma^2}$ as $\beta_{\sigma^2}/\mu_{\sigma^2} + 1$. This reparametrization helps to prevent highly correlated estimates for $\alpha_{\sigma^2}$ and $\beta_{\sigma^2}$.

Fourth, a Gaussian prior is assumed for the per-subject mean of the criterion trajectory $\mu_{x,i}$,

$$\mu_{x,i} \sim \mathrm{N}(0, \sigma_{\mu_x}^2) \tag{6}$$

with the mean being zero and $\sigma_{\mu_x}^2$ as the variance over the per-subject $\mu_{x,i}$.

Finally, we place weakly informative priors on the parameters of the hierarchical model, $\eta = (\mu_w, \sigma_w^2, \mu_a, \sigma_a^2, \mu_{\sigma^2}, \beta_{\sigma^2}, \sigma_{\mu_x}^2)$, see the S1 Appendix for complete details. These global parameters will be inferred and iteratively updated alongside the per-subject parameters using a fully Bayesian inference algorithm, which we describe next.

## A posterior inference algorithm

We develop a fully Bayesian inference algorithm to estimate the posterior distributions of the per-subject criterion trajectory and parameters, as well as the parameters of the hierarchical prior. We use an augmented, blocked Gibbs sampling algorithm to produce samples that are asymptotically drawn from the desired posterior distribution.

The Gibbs sampling algorithm proceeds by iteratively sampling the conditional distribution of one block of variables, holding the rest fixed. However, this approach requires the closed-form expressions for the conditional distributions. This tractability can be achieved if the model is conditionally conjugate. Unfortunately, the Bernoulli likelihood is not conjugate with the Gaussian hierarchical priors for the covariate weights or the AR(1) model for the criterion trajectories. To circumvent this challenge, we employ the Pólya-gamma augmentation trick [56]. With this augmentation, we can obtain a closed-form expression for the per-subject covariate weights $w_i$ given the response, covariates, criterion fluctuations, and the hierarchical priors.

For the three other per-subject parameters, $a_i$, $\sigma_i^2$, and $\mu_{x,i}$ closed-form expressions for the conditional posteriors are available. First, for $a_i$ the truncated normal prior is conjugate with the AR(1) model likelihood. Similarly, the inverse gamma prior for $\sigma_i^2$ is conjugate with the AR(1) model of the latent criterion fluctuations and the Gaussian prior is conjugate with the per-subject criterion trajectory average $\mu_{x,i}$.

After Pólya-gamma augmentation, the conditional distribution of the latent criterion fluctuations is a chain-structured Gaussian graphical model. We generate exact samples from this conditional distribution using a forward filtering/backward sampling algorithm [57]. The forward filtering step computes the filtering distributions of the latent criterion fluctuations, given all the responses up to a certain trial. Once the filtering distributions have been computed for each trial, the backward sampling step generates a sequence of latent fluctuations, working backward from the last trial $T_i$ down to the first. This procedure produces a sequence of latent fluctuations $\boldsymbol{x}_i = (x_{i,1}, \dots, x_{i,T_i})$ that is distributed according to its conditional distribution given the per-subject parameters, covariates, responses, and Pólya-gamma augmentation variables.

Finally, we update the parameters of the hierarchical prior $\eta$ by sampling from their conditional distribution under a weakly informative prior. For the three group-level parameters of the normal priors on the covariate weights and criterion trajectory mean, $\mu_w$, $\sigma_w^2$ and $\sigma_{\mu_x}^2$, we have closed-form expressions for the conditional distributions. We update the remaining

hierarchical prior parameters—$\mu_a$, $\sigma_a^2$, $\mu_{\sigma^2}$ and $\beta_{\sigma^2}$—using random-walk Metropolis-Hastings with a symmetric Gaussian proposal distribution. The parameter $\alpha_{\sigma^2}$ is then derived as $\beta_{\sigma^2}/\mu_{\sigma^2} + 1$.

Together, these updates yield a Markov chain whose stationary distribution is the posterior distribution over the latent per-subject fluctuations, per-subject parameters, and hierarchical prior parameters. Repeatedly applying these updates yields samples that are asymptotically distributed according to this posterior. We can use these samples to estimate posterior expectations and perform hypothesis tests, as demonstrated in the results below. Complete details of the augmented, blocked Gibbs sampling algorithm can be found in the S1 Appendix.

### Accessing the model

Our modeling and inference code, together with a comprehensive demonstration, is available at www.github.com/robinvloeberghs/hMFC. In addition, more in-depth plots for the simulation results are also available at this link.

## Results

To assess the effectiveness of the hMFC parameter estimation, we simulate data with a varying number of trials (500, 1000, 2500 or 5000 trials) per subject, with 50 subjects per dataset and 50 datasets per trial count. For each subject $i$, three weights, $\boldsymbol{w}_i = (w_{i,0}, \ldots, w_{i,2})$, are independently drawn from a normal distribution with mean $\boldsymbol{\mu}_w = (0.0, 0.2, -0.1)$ and variance $\sigma_w^2 = (1, 1, 1)$. Similarly, the covariates $\boldsymbol{u}_t \in \mathbb{R}^3$ are also sampled from a standard normal distribution (mean zero, identity covariance). The criterion fluctuations are sampled from an AR(1) model with the autoregressive coefficient $a_i$ drawn from a truncated normal (mean 0.98, standard deviation 0.03, minimum value 0, and maximum value 1), $\sigma_i^2$ sampled from an inverse gamma distribution ($\mu_{\sigma^2} = 0.2$, $\beta_{\sigma^2} = 0.5$), and $\mu_{x,i}$ sampled from a normal distribution with mean 0 and standard deviation 0.5.

### Group-level (global) parameter recovery

The group-level (global) parameters $\eta = (\mu_w, \sigma_w^2, \mu_a, \sigma_a^2, \mu_{\sigma^2}, \beta_{\sigma^2}, \sigma_{\mu_x}^2)$ are accurately recovered. For at least 46/50 datasets, the generative value lies within the 95% credible interval of the posterior estimates (S3 Fig). Only $\beta_{\sigma^2}$ was more difficult to recover at lower trial counts (40/50 datasets), but this did not impact the recovery of the criterion trajectories (as shown later). The recovery of the parameter improved with higher trial numbers (48/50 datasets). Most important, for $\mu_w$ and $\sigma_w^2$ the 95% credible interval contains the true value for 50/50 datasets. Note that the width of these posteriors stays relatively constant for different trial counts (S4 Fig). On the other hand, the posterior width shrinks with an increasing number of subjects. Fig 3A demonstrates, using example datasets, that increasing the number of subjects while keeping the number of trials constant (n=500) results in narrower posterior distributions. This effect is further illustrated in Fig 3B, which displays the overlaid posteriors for all 50 simulated datasets. Jointly, this suggests that in order to make more reliable inferences at the group level, for hMFC it is more advisable to gather more subjects rather then gathering more trials per subject. No problematic correlations between the inferred values of different parameters were observed, suggesting that the model can distinguish all parameters (S5 Fig). Lastly, the autocorrelation of the posterior samples quickly decays over lags, suggesting good mixing of the samples (S6A Fig).

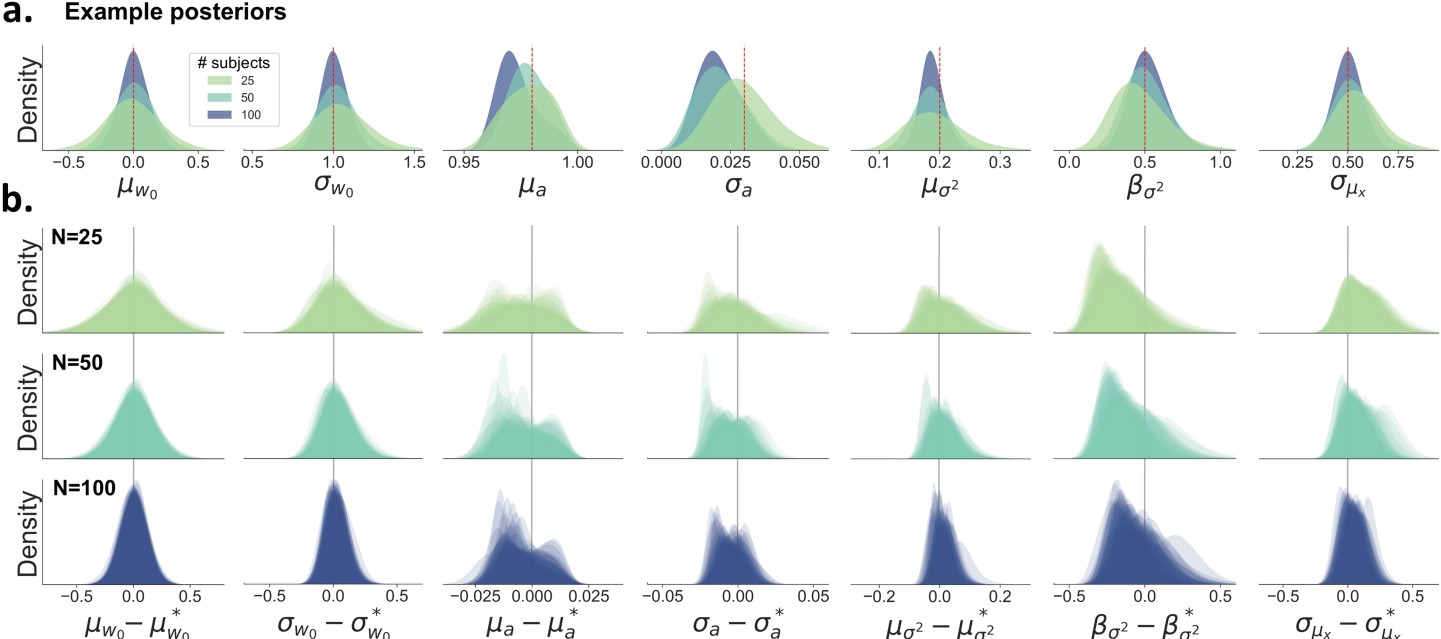

**Fig 3. An overview of the estimated posteriors distributions for the group-level (global) parameters when varying the number of subjects per dataset, with 500 trials per subject. A)** The posterior distributions become more narrow as the number of subjects increases. The true parameter values are indicated by the red dashed line. **B)** Each row shows the overlaid posteriors for all 50 simulated datasets with a varying number of subjects per dataset. The estimated posteriors are corrected and centered on the true value (denoted by an asterisk). For $\mu_w^*$ and $\sigma_w^*$ the true value is defined as the mean or standard deviation of the true per-subject parameters $w_i$ within each dataset. Due to boundaries in the parameter space, with $\sigma_w$, $\sigma_a$, $\mu_{\sigma^2}$, and $\beta_{\sigma^2}$ being strictly positive, the posterior distributions can be skewed. It should be noted that there is a slight underestimation for $\mu_a$, $\sigma_a$, and $\beta_{\sigma^2}$. This is presumably due to the true $\mu_a$ being so close to the upper boundary of 1, which causes the posterior distribution of this parameter to be asymmetric and possibly leading to compensatory effects for the other parameter estimates. Overall, we see an excellent recovery of the group-level (global) parameters.

## Per-subject (local) parameter recovery

The inference algorithm can generally recover all per-subject parameters (Fig 4A). The per-subject weights $w_{i,0}$, $w_{i,1}$, $w_{i,2}$ show excellent recovery ($r > .98$), even for datasets with only 500 trials per subject (Fig 4D). The autoregressive model parameters $a_i$, $\sigma_i^2$, and $\mu_{x,i}$, which capture the dynamics of the criterion fluctuations, show an excellent recovery with many trials per subject ($r = .94$ for $a_i$, $r = .92$ for $\sigma_i^2$, $r = .90$ for $\mu_{x,i}$ with 5000 trials), but are less accurately estimated when fewer trials per subject are available ($r = .70$ for $a_i$ and $r = .68$ for $\sigma_i^2$, $r = .63$ for $\mu_{x,i}$ with 500 trials) (Fig 4B, 4C). The autocorrelation of the posterior samples looks good (S6B Fig) and no systematic correlations were present across the true and inferred values of different parameters (S7 Fig).

## Per-subject criterion trajectory recovery

Importantly, although the per-subject parameters $a_i$, $\sigma_i^2$, and $\mu_{x,i}$ show somewhat lower recovery with low trial numbers, the model can still very accurately recover the trial-to-trial latent criterion trajectory (Fig 5) crucial for investigations into its neural and physiological bases. The recovery, measured as the correlations between the true and inferred latent trajectory, is generally high ($r > .83$), and consistent across datasets (Fig 5A). As shown for one example subject, the true and the inferred trajectory strongly correlate ($r = .88$), even though the model is not always able to perfectly recover high-frequency fluctuations in the true latent trajectory (Fig 5C).

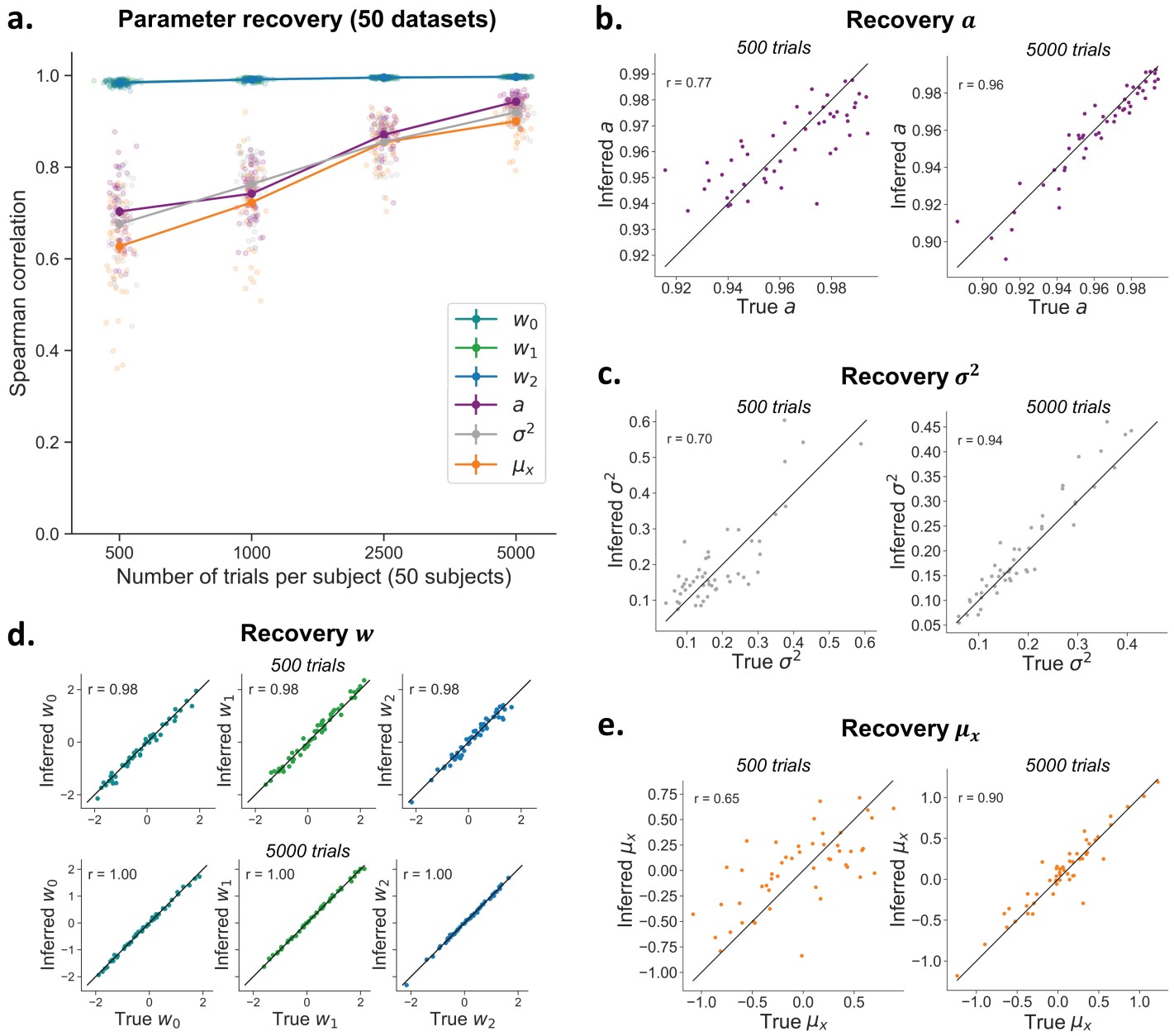

**Fig 4. Recovery of the per-subject parameters of the hMFC model. A)** The covariate weights $w$ show excellent recovery even with low trial numbers (note that the lines for the different weights overlap). The model is able to recover $a_i$ and $\sigma_i^2$ very well with high trial numbers, but recovery drops with a limited number of data points per subject. Each dot represents the recovery correlation for one dataset. Note that error bars (standard error) are shown but they are very small. **B-E)** The recovery for $a_i$ (B) and $\sigma_i^2$ (C) and $\mu_{x,i}$ (D) and covariate weights $w_{i,0}, w_{i,1}, w_{i,2}$ (E) is shown for an example dataset with 500 trials and 5000 trials per subject. Each dot represents one subject. The line shows the diagonal.

At lower trial counts (n=500), recovery ranges from 0.59 to 0.96. across subjects (Fig 5B). This variation is partly due to the stochastic criterion fluctuations themselves, but also relates to the combination of parameters governing the individual AR(1) processes. Specifically, trajectory recovery is more difficult for lower values of $a_i$ combined with higher values of $\sigma_i^2$ (S8 Fig).

**Recovery of criterion trajectory**

**Fig 5. Recovery of the latent criterion trajectory. A)** Each dot represents the recovery correlation of the criterion trajectory of one dataset, averaged over subjects. Note that error bars (standard error) are plotted but are very small. **B)** A representative dataset of 50 subjects with 500 trials each shows an average correlation between true and inferred criterion fluctuations of $r = .84$. **C)** True and inferred criterion fluctuations for an example subject. The shaded area represents the 95% credible interval of the estimated posterior at each trial.

## Computation time

Fitting the data of 50 subjects with 500 trials each for 1000 iterations takes around 1 hour on a regular laptop with only CPU's. When GPU's are available the computation time is reduced to around 5 minutes, making this procedure sufficiently lightweight to run easily on typical human datasets even for researchers without dedicated computing facilities.

## Application hMFC to sequential effects and measures of sensitivity

Having verified that the inference procedure recovers the generative hMFC model, we assess if it also corrects for the spurious effects that can arise from ignoring criterion fluctuations: apparent sequential effects and underestimated $d'$ (as shown in Fig 1). We fit the hMFC model with stimulus evidence and previous response as covariates, similar to the data shown in Fig 1 but now with $w_{\text{previous response}} = 0.5$. As a consequence, there is a repetition effect even when no criterion fluctuations are present (S9 Fig). As the criterion fluctuations become stronger, this repetition effect increases. Importantly, we ran the fits either with or without estimating criterion fluctuations, allowing us to test if the model can accurately distinguish between these two sources underlying the repetition effect.

When hMFC estimates criterion fluctuations, the true value of the previous response slope ($w_{\text{previous response}} = 0.5$, red line) falls within the posterior distribution across all conditions—even when the data contain no actual criterion fluctuations (Fig 6A). However, when the fluctuations are present but hMFC does not estimate them (i.e., assumes a fixed criterion) the repetition effects driven by the criterion fluctuations are absorbed by the previous response slope, resulting in an inflated estimate for $w_{\text{previous response}}$. As the criterion fluctuations become stronger the posterior distributions are shifted further away from the true value (i.e., replicating the finding from Fig 1C, using our model fit). These results show that not estimating criterion fluctuations while they are present leads to inflated estimates for the systematic effect of previous response. Most importantly, it also shows that hMFC can correctly

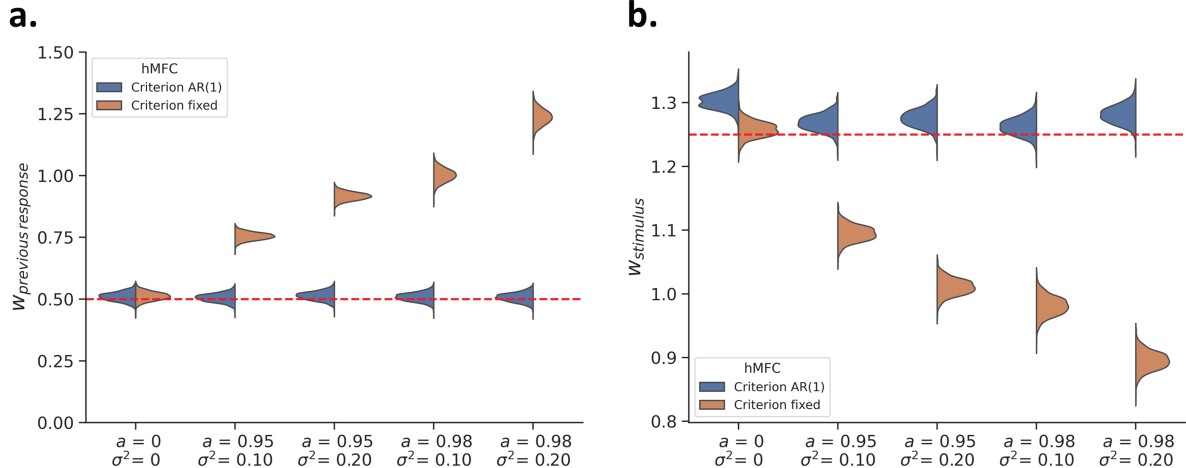

**Fig 6. The recovered posterior distributions from hMFC with the criterion estimated as AR(1) or assumed to be fixed (i.e., no fluctuations are estimated).** When criterion fluctuations are present but not estimated the posterior estimates for $w_{previous\ response}$ are inflated (A) whereas for $w_{stimulus}$ they are underestimated (B). In contrast, when the criterion fluctuations are accounted for with hMFC we see a correct recovery of the posterior estimates for both parameters. Estimating criterion fluctuations with hMFC therefore helps to address biases in these parameters.

distinguish between the two sources of choice history bias, namely a systematic effect of previous response and criterion fluctuations.

In a similar vein, when criterion fluctuations are not estimated, the psychometric slopes are systematically underestimated (Fig 6B) compared to the generative value ($w_{stimulus}$ = 1.25, red line). Fortunately, estimating criterion fluctuations resolves this underestimation. In sum, the hMFC corrects for estimation biases by explicitly estimating the fluctuations in the criterion.

## Applying hMFC to an empirical dataset

In the previous sections we have shown that hMFC accurately recovers the generative parameters on simulated data, and that it corrects for biases that occur when not measuring criterion fluctuations. As a final endeavor, we effectively fitted hMFC to an empirical data by Shekhar and Rahnev (2021) [58], consisting of 20 subjects with 2790 trials each. In addition, using these empirical fits we will demonstrate the benefit of having the autoregressive coefficient $a_i$ as a free parameter, instead of assuming a fixed value [21,23] by simulating data based on the empirical parameter estimates.

Estimating hMFC to the empirical dataset reveals values ranging from 0.81 to 0.96 for $a_i$, and values between 0.05 and 0.35 for $\sigma_i^2$ (Fig 7A). Using the per-subject parameter estimates and the empirical data we simulated a new dataset and fitted two versions of hMFC in which $a_i$ is freely estimated or in which $a_i$ is fixed to .9995. As can be seen in Fig 7B–7C hMFC is able to recover the parameter well when $a_i$ is freely estimated. However, when $a_i$ is fixed both the recovery of the parameters and the criterion trajectories suffer. In fact, Fig 7D shows that when $a_i$ is fixed the correlation between the true and inferred criterion trajectory is consistently lower ($t = 7.37$, $p < 0.001$) and the root mean square error (RMSE) higher ($t = -8.30$, $p < 0.001$). These results show that hMFC is able to recover the parameter range found in empirical data, and demonstrate the importance of having $a_i$ as a free parameter.

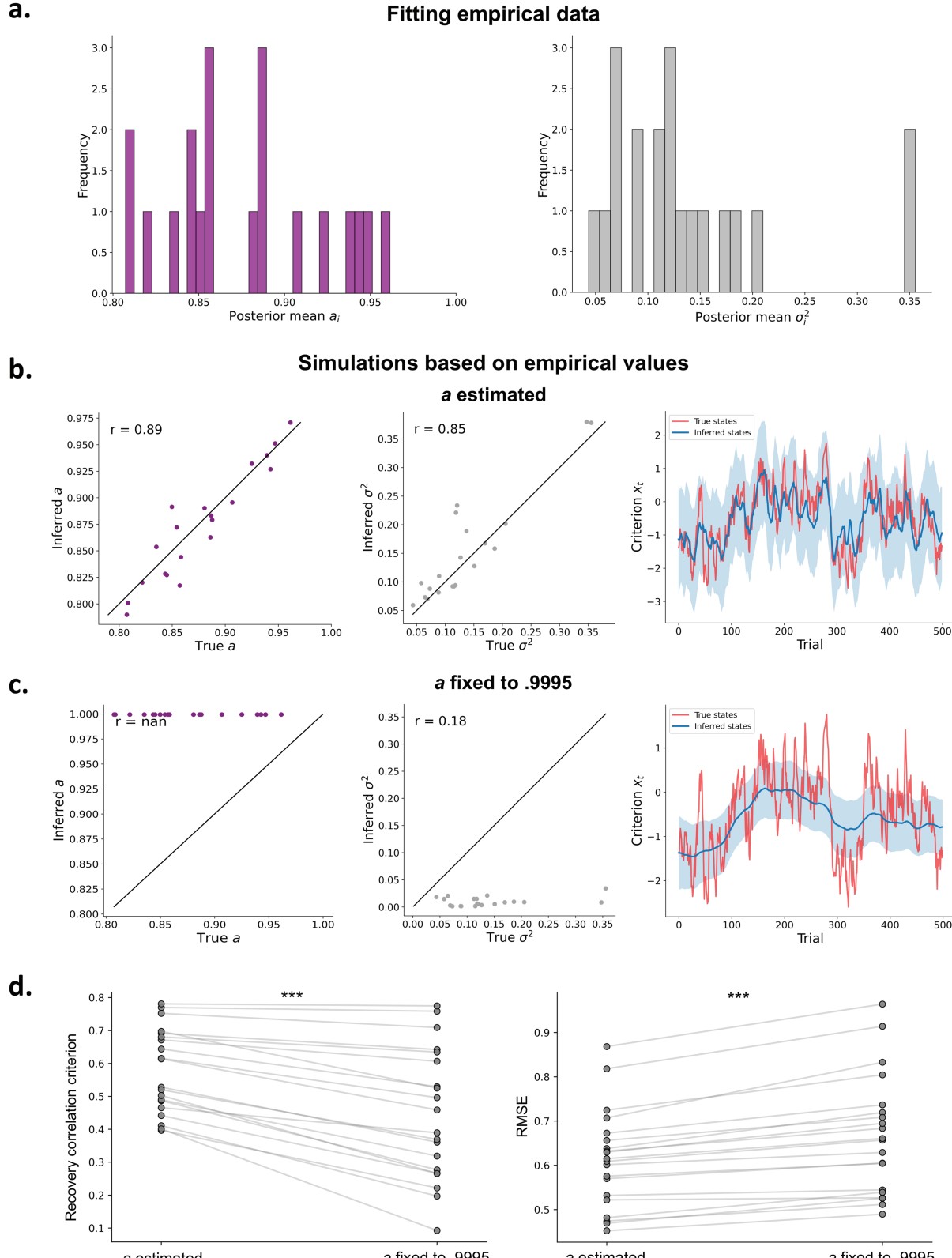

**Fig 7. Applying hMFC to an empirical dataset. A)** The estimated posterior means for each subject of $a_i$ and $\sigma_i^2$ when fitting hMFC to an empirical dataset of Shekhar and Rahnev (2021) [58]. Based on these empirical fits of the per-subject parameters we simulated a new dataset and refitted two variants of hMFC where $a_i$ was freely estimated or fixed to .9995 (following Gupta and Brody (2022) [21], Roy et al. (2021) [23]). **B)** When $a_i$ is freely estimated we see a good recovery for the parameters $a_i$ and $\sigma_i^2$, and the criterion trajectory

(example subject with first 500 trials shown). **C)** With $a_i$ fixed to .9995 we see a systematic underestimation of $\sigma_i^2$. Most importantly, this also affects the recoverability of the criterion trajectory. **D)** For all 20 subjects the correlation between the true and estimated criterion trajectory is higher with $a_i$ as a free parameter compared to having it fixed. Similarly, the root mean squared error (RMSE) is lower when $a_i$ is estimated. However, accurate recovery of $a_i$ at low trial counts requires a hierarchical estimation procedure (S1 Fig).

## Discussion

Computational models of decision-making typically assume stationarity in the parameters underlying the decision computation. However, an increasing body of research highlights the dynamic nature of decision making and shows that computational variables underlying decision-making may fluctuate over time. One example of these non-stationarities are fluctuations in the decision criterion. Ignoring such latent fluctuations can lead to false interpretations and inaccurate estimates. Specifically, criterion fluctuations can lead to behavior that mimics a causal relationship in sequential effects, and they can lead to an underestimation of two popular measures of perceptual sensitivity ($d'$ in SDT and the psychometric slope).

Here, we developed the Hierarchical Model for Fluctuations in Criterion (hMFC), a Bayesian hierarchical model to obtain trial-by-trial estimates of the decision criterion. Even with only 500 trials per participant, which is common in studies on human cognitive neuroscience, the model parameters and the latent criterion trajectory itself can be accurately inferred. Importantly, the hMFC can effectively correct for biases that occur when criterion fluctuations are ignored.

Although hMFC confers several advantages, it is not without limitations. A first limitation is that it only allows an AR(1) process to govern the decision criterion, i.e., the intercept of the model, whereas it assumes all other parameters are fixed. Future work may explore how to extend this hierarchical Bayesian framework to simultaneously infer fluctuations in the weights of other covariates, e.g. those reflecting stimulus sensitivity or trial history [23]. A second limitation is that hMFC uses observed choices to estimate latent trajectories, but it does not consider reaction times. Future work may explore how to extend this approach to also incorporate reaction times in the estimation of the latent states of parameters. Finally, as showcased in the paper hMFC works very well for estimating the latent trajectory of the decision criterion. Therefore, researchers seeking to infer those trajectories in their datasets will find hMFC a useful tool. Researchers that are, however, interested in the parameters giving rise to these fluctuations (e.g., $a_i$ and $\sigma_i^2$) should carefully inspect the parameter recovery results to determine in which cases they can interpret these estimates. Indeed, for very low trial counts or low number of participants some group-level parameter estimates are not always within the 95% credibility interval of the true values. In those cases, focusing on acquiring data from subjects or more trials per subject could effectively alleviate these concerns.

We provide open access to the code for hMFC and created an interactive demo for researchers interested in applying this model to their data. Through this work, we hope to enable further research in human cognitive neuroscience aimed at understanding the dynamics of decision-making.

## Supporting information

**S1 Appendix. Detailed mathematical description of Hierarchical Model for Fluctuations in Criterion (hMFC), including notation, model specification, prior distributions, posterior inference with Gibbs sampling, and initialization procedures.**
(PDF)

**S1 Fig. Recovery of a fluctuating decision criterion using traditional methods.**
(TIF)

**S2 Fig. Recovery of per-subject AR(1) parameters in a simulated dataset with 50 subjects, illustrating the benefits of hierarchical priors when trial counts are low.**
(TIF)

**S3 Fig. Overview of frequency with which true parameter values fall within 95% credible intervals as a function of subject and trial counts.**
(TIF)

**S4 Fig. Recovery of group-level parameters across varying numbers of trials per subject.**
Example posteriors and overlaid posterior distributions are shown.
(TIF)

**S5 Fig. Correlations between inferred group-level parameters across 50 simulated datasets.**
(TIF)

**S6 Fig. Autocorrelation plots for group-level and per-subject parameters.**
(TIF)

**S7 Fig. Correlations between true and inferred per-subject parameters.**
(TIF)

**S8 Fig. Correlation between true and estimated criterion trajectories as a function of the true AR(1) parameters ($a$ and $\sigma^2$).**
(TIF)

**S9 Fig. Effect of criterion fluctuations on response repetition behavior in simulated data with a systematic effect of previous response.**
(TIF)

## Acknowledgments

We thank Diksha Gupta, Peter Dayan, Hans op de Beeck, and Stijn Verdonck for the technical discussions and useful feedback on the project.

## Author contributions

**Conceptualization:** Robin Vloeberghs, Anne E. Urai, Kobe Desender, Scott W. Linderman.

**Formal analysis:** Robin Vloeberghs, Scott W. Linderman.

**Funding acquisition:** Anne E. Urai, Kobe Desender, Scott W. Linderman.

**Investigation:** Robin Vloeberghs, Kobe Desender.

**Methodology:** Robin Vloeberghs, Scott W. Linderman.

**Project administration:** Robin Vloeberghs.

**Software:** Robin Vloeberghs, Anne E. Urai, Scott W. Linderman.

**Supervision:** Anne E. Urai, Kobe Desender, Scott W. Linderman.

**Visualization:** Robin Vloeberghs.

**Writing – original draft:** Robin Vloeberghs, Kobe Desender, Scott W. Linderman.

**Writing – review & editing:** Robin Vloeberghs, Anne E. Urai, Kobe Desender, Scott W. Linderman.

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
