## [Decision Letter · Decision Letter 0]

29 Jan 2025

 PCOMPBIOL-D-24-01687

A Bayesian Hierarchical Model of Trial-To-Trial Fluctuations in Decision Criterion

PLOS Computational Biology

Dear Dr. Vloeberghs,

Thank you for submitting your manuscript to PLOS Computational Biology. After careful consideration, we feel that it has merit but does not fully meet PLOS Computational Biology's publication criteria as it currently stands. Therefore, we invite you to submit a revised version of the manuscript that addresses the points raised during the review process.

While the reviewers are generally very positive about the manuscript, there are still some questions about the presentation of the advantages of the model (see especially reviewer 1).

Please submit your revised manuscript within 60 days Mar 31 2025 11:59PM. If you will need more time than this to complete your revisions, please reply to this message or contact the journal office at ploscompbiol@plos.org. Please include the following items when submitting your revised manuscript:

We look forward to receiving your revised manuscript.

Kind regards,

Ulrik R. Beierholm

Academic Editor

PLOS Computational Biology

Andrea E. Martin

Section Editor

PLOS Computational Biology

**Journal Requirements:**

3) Your manuscript is missing the following section: Methods.  Please ensure all required sections are present and in the correct order. Make sure section heading levels are clearly indicated in the manuscript text, and limit sub-sections to 3 heading levels. An outline of the required sections can be consulted in our submission guidelines here:

5) We notice that your supplementary Figures and information are included in the manuscript file. Please remove them and upload them with the file type 'Supporting Information'. Please ensure that each Supporting Information file has a legend listed in the manuscript after the references list.

6) We note that your Data Availability Statement is currently as follows: "All relevant data are within the manuscript and its Supporting Information files." Please confirm at this time whether or not your submission contains all raw data required to replicate the results of your study. Authors must share the “minimal data set” for their submission. PLOS defines the minimal data set to consist of the data required to replicate all study findings reported in the article, as well as related metadata and methods (https://journals.plos.org/plosone/s/data-availability#loc-minimal-data-set-definition).

7) Please amend your detailed Financial Disclosure statement. This is published with the article. It must therefore be completed in full sentences and contain the exact wording you wish to be published.

1) Please state: "The funders had no role in study design, data collection and analysis, decision to publish, or preparation of the manuscript.".

**Reviewers' comments:**

Reviewer's Responses to Questions

Reviewer #1: In this manuscript, Vloeberghs et al. describe a hierarchical model that tracks trial-to-trial fluctuations in decision criterion. The paper's contributions include code showing how their model can be applied by other researchers.

The paper tackles an interesting and relevant problem, and their methods will be of interest to many researchers. However a number of the paper’s main claims about the novelty/usefulness of the authors’ methods warrant more evidence and/or more precise wording, especially in light of past work. Additionally, the provided demos and discussion warrant some restructuring for the sake of readability and usefulness. I have detailed these points below:

Major comments

Accuracy of estimating fluctuations with lower data requirements

One of the key claims about the novelty and benefits of the authors’ methods, is that it has lower data requirements than previous methods, as it accurately recovers parameter and criterion estimates with as little as 500 trials. However, this claim needs to be qualified for the following reasons:

In figure 3, we see that estimated parameters of the criterion drift (mu_a, sigma_a, alpha and beta) show a systematic bias - they are all systematically underestimated even with increasing numbers of subjects - the authors do not address this bias in recovery. The authors claim that recovered estimate lies in the 95% credible intervals but these intervals are not shown.

In figure 4a, we see that the spearman correlation of recovered parameters drops off significantly (close to 60%) for some of the parameters when trial counts reach 500 trials. Relatedly, both Roy et al and Gupta & Brody do in fact fit datasets on the order of 1000 trials (Fig 7 in Roy et al, Fig 3E of Gupta & Brody) showing similar levels of recovery at such low trial counts.

Due to these issues, it is unclear what performance advantage the hMFC is buying at these low trial counts compared to previous techniques - perhaps the authors could show the advantage of fitting with v.s. without hierarchical priors to show what the hierarchical nature of the model adds.

Benefits of estimating rather than fixing certain parameters

The authors note that in their model, they treat the autoregressive coefficient “a” as a free parameter, rather than fixing it at 1 or 0.9995 as Roy et al and Gupta & Brody have done. However it is unclear how beneficial this actually is:

In fig 4b, one can see that very few different values of “a” are actually tested, within a narrow range of 0.94-1. This is probably owing to the fact that as shown in Supplementary Figure S3, the correlation between true and estimated trajectory drops off quickly as “a” values decrease below 0.99.

As mentioned before, recovery of “a” values at 500 trials is quite poor (~60%) however, from Fig 5a the recovery of the trajectory doesn’t suffer due to this misestimation. This is likely because the process of smoothing to estimate latent trajectories is tolerant to slight mis-specification of “a”, hence it is unclear how much value we are getting from estimating it as a free parameter.

Restructuring required in discussion/demos

The provided code demos seem to be a collection of .py python files (presumably converted from .ipynb Jupyter notebooks) - this format is not interactive, and does not allow other researchers to easily run individual sections, replace with their own datasets or compare their output to reference outputs from the authors’ runs. For the sake of reproducibility and ease of use, I suggest that the authors provide the original Jupyter notebooks (.ipynb) showing the expected outputs of each cell and allowing anyone to easily interact with their demos.

The discussion is currently a bit impoverished, moreover a number of sections currently in the introduction (everything from line 80: “Why should we care about fluctuations in decision criterion?” onwards until the methods) are more suited to the discussion as they discuss significance of the authors’ methods/results w.r.t previous work, and I would suggest they be moved to the discussion.

Minor comments:

Line 214-217, 235 - 237: Unclear what this is referring to, since the parameterization in Gupta and Brody does not seem to have any parameter weighing the criterion fluctuations

The authors sometime refer to the drifting latent criterion as a “latent state” when describing the model - better to use consistent terminology and refer to the drifting criterion everywhere.

Include citation to dynamax

Legends in s3 are not matched to the plotted range

Fig 6 legend criterion estimated v.s. not estimated -> criterion is AR(1) v.s. criterion is fixed

Reviewer #2: In this paper, Vloeberghs and colleagues propose a model (and provide the corresponding Python code on a github repository) for estimating autocorrelated fluctuations in the decision criterion in a two-alternative forced-choice task. The paper begins by presenting the unfortunate consequences of ignoring such fluctuations. When not taken into account, fluctuations in the decision criterion are mistaken for reduced sensitivity (psychometric slope, d') and sequential effects (with an apparently high probability of repeating the previous choice). The paper shows (at the end) that when random drift in the decision criterion is taken into account, the estimates of sequential effects and sensitivity are no longer biased. The proposed model differs from previous models in that it explicitly models the random fluctuations in the criterion (unlike Lak et al 2020), it estimates the temporal dependence of the autocorrelation (unlike Roy et al 2021), and it uses a hierarchical fitting approach to regularise the estimates in the absence of a large number of trials (unlike what is done e.g. with rodents, Gupta & Brody 22).

I have looked at (but not tested) the code. It appears to be well written and documented. It also seems to be fast because it is written using jax, which makes it possible to benefit from faster computation time when GPUs are available, without the user having to adapt the code.

The appendix explains the mathematics implemented in the code very clearly, it's easy to follow.

Overall, I think this methodological/software paper is useful and very well written. It draws the community's attention to the problem of variability in the decision criterion and its consequences, and proposes a solution that is applicable in typical human experiments (where the number of trials is typically limited) and is readily available in Python in a nicely documented repository. The paper presents a number of quality checks for their methods and code.

I have only one request regarding the parameter recovery presented in Fig. 3 and Fig. 4. One aspect is missing: we would also like to know to what extent the parameters can be distinguished from each other by reporting the correlations between the fitted values of parameter k and the true value of parameter j (and not only the special case j=k presented in Fig. 4). This is particularly interesting for parameters a and \sigma of the autocorrelation model.

Reviewer #3: This paper introduces a hierarchical Bayesian model for estimating parameters that capture fluctuations in the decision criterion. The model effectively estimates these parameters at both individual and population levels, even with limited data. The authors demonstrate that neglecting fluctuating decision criteria can lead to misinterpretations, such as attributing data to choice history biases or underestimating perceptual sensitivity.

With growing recognition of the importance of understanding and accounting for fluctuations in decision criteria, the ability to do so with fewer trials makes this approach highly appealing to a broader community of researchers interested in both behavioral and neural mechanisms of decision-making.

The paper is very well written with detailed explanations of the potential mis-interpretations that can happen when fluctuating criterion is not accounted for. The results are rigorous and figures are easy to understand.

However, I do have a few small questions:

— the authors claim that not accounting for decision fluctuations can lead to misinterpretations of the data (fig 1c-e). However, my concern is, if a subject does indeed have choice history bias, fitting this model might incorrectly attribute that to fluctuating decision criterion. Can this model be extended to account for both? Or it is for researchers using this tool to decide if they want to fit two separate models and do model comparison to know the true underlying mechanism.

— Additionally, it would add value to see if the authors can use some of the available empirical data in literature to fit this model and explain the observations.

— Since the authors are using Gibbs sampling, I was wondering if they could highlight some parameter ranges where the algorithm would not be able to converge with limited iterations due to correlation of variables

— Finally, it would help to discuss some limitations of the model/tool, maybe the top few issues that researchers using the tool should be aware of.

**Have the authors made all data and (if applicable) computational code underlying the findings in their manuscript fully available?**

Reviewer #1: Yes

Reviewer #2: Yes

Reviewer #3: Yes

PLOS authors have the option to publish the peer review history of their article (what does this mean?). If published, this will include your full peer review and any attached files.

Reviewer #1: No

Reviewer #2: No

Reviewer #3: No

**Figure resubmission:**
---

## [Decision Letter · Decision Letter 1]

2 Jul 2025

Dear Vloeberghs,

We are pleased to inform you that your manuscript 'A Bayesian Hierarchical Model of Trial-To-Trial Fluctuations in Decision Criterion' has been provisionally accepted for publication in PLOS Computational Biology.

Best regards,

Ulrik R. Beierholm

Academic Editor

PLOS Computational Biology

Andrea E. Martin

Section Editor

PLOS Computational Biology

Reviewer's Responses to Questions

**Comments to the Authors:**

Reviewer #1: I appreciate the authors' thoughtful and thorough responses to my questions. I have no further concerns and support publication of the results.

Reviewer #2: the authors have responded satisfactorily to my concern.

Reviewer #3: The authors have thoroughly addressed the reviewers' comments, including my own, through substantial revisions that include new figures, expanded analyses, and relevant updates to the main text.

One of my main concerns was whether the model could dissociate between fluctuating decision criteria and choice biases. The authors have convincingly clarified that the model can account for both. They provide a detailed explanation, supported by additional analysis in Figure S9 and Figure 6, which demonstrate that when criterion fluctuations are present but not explicitly estimated, the posterior estimates for _previous_response become inflated—thereby validating the model's sensitivity to this distinction.

Another point I raised was the need to validate the model against empirical data. The authors have addressed this by fitting the model to a dataset from Shekhar & Rahnev (2021), which includes 20 participants with 2,790 trials each. This analysis, now presented in Figure 7, shows that the hMFC model successfully recovered parameters from real behavioral data.

To address concerns regarding potential issues with parameter mixing when using the Gibbs sampling method, the authors conducted simulations across a wide range of parameters. They evaluated autocorrelations among inferred values and found no problematic correlations, demonstrating that their inference procedure is robust for their application.

Finally, the revised manuscript includes a thorough and thoughtful discussion of the model's limitations, which enhances the overall transparency and credibility of the work.

**Have the authors made all data and (if applicable) computational code underlying the findings in their manuscript fully available?**

Reviewer #1: Yes

Reviewer #2: Yes

Reviewer #3: Yes

PLOS authors have the option to publish the peer review history of their article (what does this mean?). If published, this will include your full peer review and any attached files.

Reviewer #1: No

Reviewer #2: No

Reviewer #3: No

---

## [Editor Report · Acceptance letter]

PCOMPBIOL-D-24-01687R1

A Bayesian Hierarchical Model of Trial-To-Trial Fluctuations in Decision Criterion

Dear Dr Vloeberghs,

I am pleased to inform you that your manuscript has been formally accepted for publication in PLOS Computational Biology. Your manuscript is now with our production department and you will be notified of the publication date in due course.

With kind regards,

Zsofia Freund
